# Using mobile health technology and community health workers to identify and refer caesarean-related surgical site infections in rural Rwanda: a randomised controlled trial protocol

Kristin A Sonderman,[1,2] Theoneste Nkurunziza,[3] Fredrick Kateera,[3] Magdalena Gruendl,[2] Rachel Koch,[2] Erick Gaju,[4] Caste Habiyakare,[4] Alexi Matousek,[1] Evrard Nahimana,[3] Georges Ntakiyiruta,[5] Robert Riviello,[1,2] Bethany L Hedt-Gauthier[2,3]

[1]Center for Surgery and Public Health, Brigham and Women's Hospital, Boston, Massachusetts, USA
[2]Department of Global Health and Social Medicine, Harvard Medical School, Boston, Massachusetts, USA
[3]Partners In Health, Kigali, Rwanda
[4]Ministry of Health, Kigali, Rwanda
[5]Ejo Heza Surgical Centre, Kigali, Rwanda

**Correspondence to**
Dr Kristin A Sonderman; ksonderman@bwh.harvard.edu

## ABSTRACT

**Introduction** Surgical site infections (SSIs) are a significant cause of morbidity and mortality in low-income and middle-income countries, where rates of SSIs can reach 30%. Due to limited access, there is minimal follow-up postoperatively. Community health workers (CHWs) have not yet been used for surgical patients in most settings. Advancements in telecommunication create an opportunity for mobile health (mHealth) tools to support CHWs. We aim to evaluate the use of mHealth technology to aid CHWs in identification of SSIs and promote referral of patients back to healthcare facilities.

**Methods and analysis** Prospective randomised controlled trial conducted at Kirehe District Hospital, Rwanda, from November 2017 to November 2018. Patients ≥18 years who undergo caesarean section are eligible. Non-residents of Kirehe District or patients who remain in hospital >10 days postoperatively will be excluded. Patients will be randomised to one of three arms. For arm 1, a CHW will visit the patient's home on postoperative day 10 (±3 days) to administer an SSI screening protocol (fever, pain or purulent drainage) using an electronic tablet. For arm 2, the CHW will administer the screening protocol over the phone. For both arms 1 and 2, the CHW will refer patients who respond 'yes' to any of the questions to a health facility. For arm 3, patients will not receive follow-up care. Our primary outcome will be the impact of the mHealth-CHW intervention on the rate of return to care for patients with an SSI.

**Ethics and dissemination** The study has received ethical approval from the Rwandan National Ethics Committee and Partners Healthcare. Results will be disseminated to Kirehe District Hospital, Rwanda Ministry of Health, Rwanda Surgical Society, Partners In Health, through conferences and peer-reviewed publications.

**Trial registration number** NCT03311399.

## INTRODUCTION
### Background
Surgical site infections (SSIs) are a major source of morbidity and mortality worldwide and the leading healthcare-associated

### Strengths and limitations of this study

► The greatest strength is that this is a prospective randomised controlled trial to most effectively evaluate the impact of a mobile health and community health worker (CHW) intervention on return to care following surgery.
► The screening protocol used has been previously validated by the study team in this setting.
► The study is well-resourced with significant on the ground logistical support through Partners In Health and the staff at Kirehe District Hospital.
► In addition to assessing the impact on patient return-to-care behaviours, this study will also allow us to describe the feasibility of mobile health and CHW interventions in this setting, beyond surgical interventions.
► Since validating the presence or absence of postoperative infections would interfere with the study aims, we can only compare the proportion of all patients that return to care with confirmed infections and must assume that the infection rates across arms are constant.

infection in the low-income and middle-incoume countries (LMICs).[1] The burden is disproportionately felt in LMICs, and especially by those in Africa where the rates of postoperative SSIs have been documented as high as 30.9%.[2] In these settings, SSIs often develop after patients are discharged home, and geographic and resource barriers prevent patients from routine postoperative follow-up.[3 4] In many LMICs, including Rwanda, follow-up with a surgical care provider after a procedure is not routine. Even when scheduled, rates of follow-up are low. A study from Central African Republic reported only 25% of surgical patients returned for

their scheduled 30-day postoperative visit.[5] For patients who develop an SSI, failure to return or a delayed return to care is linked with poor health outcomes including sepsis, need for reoperation, death and increased healthcare costs.[6]

In many LMICs, community health workers (CHWs) play a major role in delivering household-based care to vulnerable populations who might otherwise be unable to access health facilities.[7 8] Globally, the range of responsibilities of CHWs vary by programme, whether polyvalent or topic-focused, such as the maternal and child health CHWs in Rwanda.[9] Regardless of the range, the number of responsibilities is typically high leading to CHW work overload. Additional activities require extensive preservice or postservice training or provision of activity support aides. Recent advances in telecommunication and increasing access to mobile phones in LMICs create opportunities to use mobile health (mHealth) strategies to support CHWs. In Rwanda, 63% of the population in 2014 reportedly owned a cell phone, with 99% having access to mobile networks.[10] Multiple studies have shown that real-time use of mHealth technologies increases adherence to health protocols in rural Africa,[11–15] and also improves the perceived quality of care.[16]

In 2014, members of the study team carried out a pilot study in Haiti that involved CHWs following up with surgical patients once discharged and evaluating their wounds for an SSI.[17] The CHWs used an mHealth application that prompted the CHW to evaluate the wound for certain characteristics pertaining to SSIs as well as to take a photograph of the wound. The CHW's assessment of the wounds were then compared with a surgeon's assessment (using the photograph), and found 85% agreement. In the phase I study precluding this manuscript, over a 4-month period in 2017 (March– July) at KDH, we evaluated caesarean section (C-section) patients at postoperative day (POD) 10 (±3 days) and found a 10.3% SSI incidence (results yet to be published). In this study, we draw from lessons learnt in the pilot to rigorously explore the use of mHealth-CHW interventions for postoperative follow-up of patients delivering via C-sections in rural Rwanda.

### Aims
The overall study aim is to examine whether CHWs, guided by an mHealth-delivered screening protocol, can improve the identification of SSIs and inform a timely return to care among patients who undergo C- sections.

Specific objectives:
► Objective 1: evaluate the impact of the mHealth-CHW interventions on patients returning to the health centre or hospital for a possible SSI.
► Objective 2: to assess the feasibility of an mHealth-CHW intervention for postoperative follow-up.

After receipt of a voluntary written consent, enrolled patients will be randomly assigned to one of three arms (figure 1):
► Arm 1: an mHealth-CHW intervention where the CHW visits the patient postoperatively, administers the screening protocol and refers the patient back to care if there is evidence of an SSI.

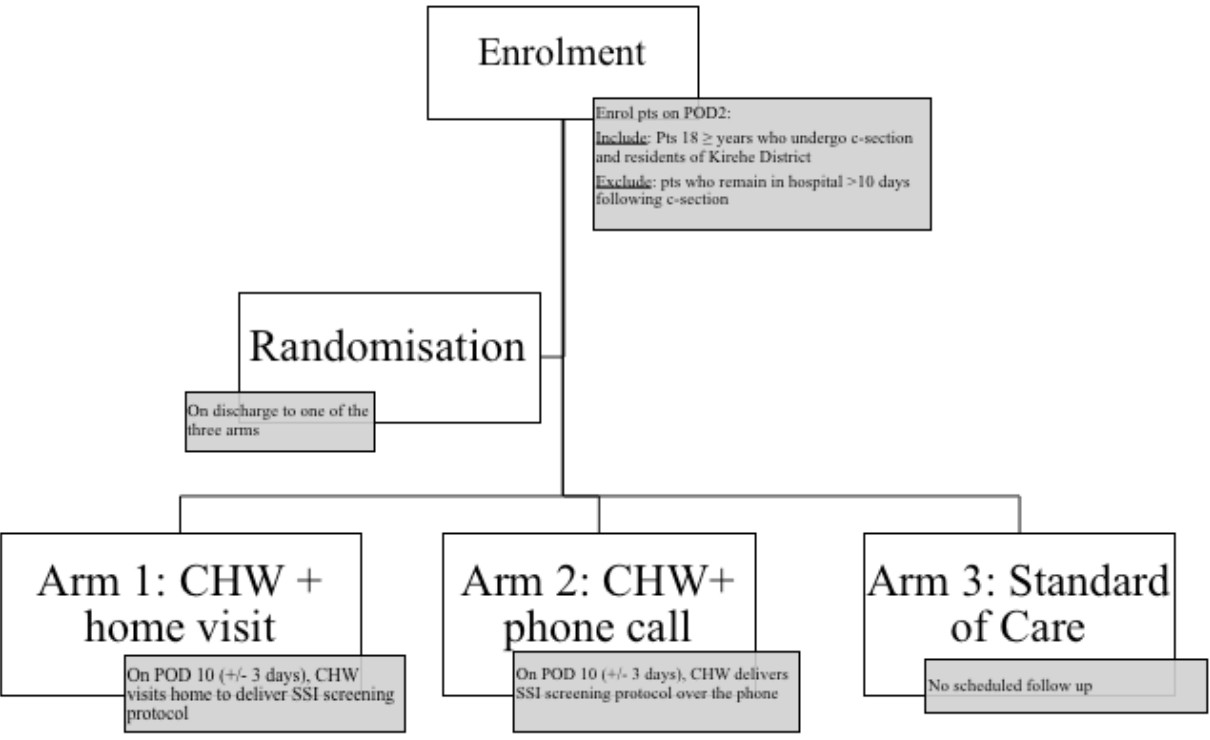

**Figure 1** Study design. CHW, community health workers; POD, post operative day; SSI, surgical site infection.

- ► Arm 2: an mHealth-CHW intervention where the CHW calls the patient postoperatively, administers the screening protocol over the phone and refers the patient back to care if there is evidence of an SSI.
- ► Arm 3: standard of care (no routine follow-up).

## METHODS AND ANALYSIS
### Study location
This study will take place between November 2017and November 2018 at Kirehe District Hospital (KDH), one of 42 district hospitals in Rwanda. KDH is a level 1 hospital, with 235 beds, operated by the Rwanda Ministry of Health and supported by the medical non-profit organisation Partners In Health (PIH). The hospital serves a catchment area of 368 950 people, primarily residing in rural, outlying villages. KDH performs around 1400 surgical operations a year, with the majority being C-sections.[18] Nearly all C-sections are performed by general practitioner (GP) physicians, with occasional surgeries performed by visiting obstetricians.

### Study population
This study will only include patients undergoing C-section delivery, which are the majority of patients undergoing an operation, at KDH. Over 60% of all surgeries performed at Rwandan district hospitals are obstetric related.[19] All patients aged 18 years or older undergoing a C-section at KDH during a 12-month study enrolment window will be eligible for inclusion. Participants must be residents of Kirehe District. We will exclude patients who remain inpatient after POD 10 as the window for follow-up we are interested in would have passed (10 days postoperative ±3 days). We will also exclude patients who are residents of Mahama refugee camp in Kirehe as the refugee camp is temporary and the patients are not covered by the existing CHW network.

### The SSI screening protocol
The study SSI screening protocol will consist of three screening questions, which were developed and validated during phase I of this study. Phase I was also carried out at KDH, and the three questions were selected to have the highest sensitivity while maintaining reasonable specificity for diagnosing an SSI. The optimisation occurred over a 4-month period in 2017 and included post-C-section surgical discharged patients aged 18 years or older. Patients returned to the hospital for evaluation on POD 10 (±3 days) and were evaluated by a GP. A CHW administered a 10-question SSI screening protocol assessing for: 1) increased pain since discharge; 2) fever since discharge, 3) erythema, 4) edema, 5) induration, 6) dehiscence, 7) drainage from the wound, 8) drainage with discolouration, 9) drainage with a foul odour and 10) drainage with pus (purulent drainage). Using the GP's SSI diagnosis as the gold standard, we identified the following three questions as most sensitive and specific for SSI diagnosis: purulent drainage, pain and fever (table 1).

**Table 1** Surgical site infection screening protocol

| Question | Answer |
| --- | --- |
| Have you had a fever since discharge? | Yes/no |
| At the incision, have you had increasing pain? | Yes/no |
| Any active drainage? | Yes/no |
| What colour is the fluid? | Brown, yellow, green or white/red, pink, clear |

### Study interventions
The study involves two different interventions: use of mHealth and CHWs arms. For both interventions, patients will be screened at POD 10 (±3 days). We selected this window because the majority of SSIs develop between POD 5 and 10 days and timely identification of SSIs is a critical aspect of the intervention.[20] In arm 1, a CHW will travel to the patient's home to evaluate the patient. Prior to the visit, the patient will be called to confirm location and time. The hired surgical CHW will contact the local CHW who will guide the surgical CHW to the patient's home. Once at the patient's home, the local CHW will leave, and the surgical CHW will evaluate the patient using the SSI screening protocol administered on an electronic tablet and take a photo of the wound. In arm 2, a CHW will call the patient on the phone on POD 10 (±3 days). Three attempts will be made to reach the patient. The CHW will administer the screening protocol over the phone, prompted by the tablet application to ask the appropriate questions. For both intervention arms, if the patient answers yes to any of the three questions, the patient will be instructed by the CHW to present to the local health centre for evaluation and referral to KDH if necessary. Patients not identified with an SSI will be reminded of proper wound care, warning signs of SSI and to follow-up should there be any change. In arm 3, patients will be given discharge instructions; however, they will not have any contact with a CHW following discharge and therefore will serve as a control group.

### Study consent, enrolment, randomisation and follow-up
On POD 2, eligible patients will be identified. Study staff will read the consent form (see online supplementary appendix) to the patient in Kinyarwanda and solicit a signed consent. Once the patient is enrolled, there will be no special retention strategies as this will interfere with the overall study outcomes.

At discharge, the enrolled patients will be randomised to one of the three study arms described above. Study staff will prepare study packets, in sealed envelopes numbered consecutively. REDCap application will be used to randomly generate arm assignments to each packet. The assignment is independent of any patient factors, including whether the patient has access to a cell phone or lives in an area with cell phone coverage. In addition to the random arm assignment, the packet will include

details on arm-specific follow-up such as follow-up plan for home visits (arm 1) or phone call date (arm 2). The packet will also include general discharge instructions including signs of a surgical site infection, how to contact study staff and how to return to a health centre for care or referral to KDH if an SSI is suspected by CHW.

All enrolled patients will be followed for 30 days postoperatively. If a patient is identified as having an SSI, she will be followed up to 90 days to document the progression and treatment of the infection. On POD 30, all patients will be called by a member of the study team to check in to see if they have returned to care. Study participants who return to care will be recorded by the register at the health facility where she presents (health centre or hospital), and the study team will have regular check-ins with the register to obtain the list of patients who returned to care. Clinical data from those follow-up visits will then be transcribed into REDCap for each patient.

## Data collection and variables

All study data will be collected, managed and store using REDCap electronic data capture tools hosted by Brigham and Women's Hospital. REDCap is a secure web application that can support both online or offline data collection for research studies.[21 22] The REDCap mobile application will be used by CHWs to administer the SSI screening protocol. There will be five distinct time points of data collection. Study coordinators will have access to data to evaluate for completeness.

First, on enrolment, all patients will provide basic demographics, socioeconomic and location data including but not limited to age, occupation, education, household income, insurance, home location, travel distance from the patient's home, patient's home village, cell, sector name, name of local CHW, phone number of the patient, phone number of a family member or a neighbour (in case the patient does not have personal phone), with permissions to call these numbers as part of follow-up. Second, on discharge, data collectors will complete a clinical chart review, extracting details on patient's past medical history, intraoperative data (preoperative antibiotics, wound class, intraoperative complications) and postoperative care. Third, for patients in arm 1 and arm 2, we will collect the responses of the SSI screening protocol. The CHW will click on the patient's ID number in REDCap, and the application will prompt the CHW to ask the three SSI screening protocol questions. The CHW will answer the questions on the tablet and the data will be stored. The fourth round will include the CHW separately collecting data on process indicators related to the implementation of the intervention. For arm 1, these indicators will include: ability to visit the patient on the scheduled date, ability to find the local CHW and the patient's home, travel time, presence of the patient at time of visit, willingness of the patient to allow CHW into home, patient compliance with the SSI screening protocol, if the patient allowed the CHW to perform an examination/take a photo of the wound and if there were any technical difficulties with the tablet or software. For arm 2, these indicators will include: whether the patient was reached by phone, how many attempts were made, which number was called and who answered, total call time and whether patient allowed the CHW to administer the SSI screening protocol. Finally, we will track the patient's return to care within 30 days postoperatively using a register posted at each of the 16 district health centres where staff can record any study patients who present to that location for care. The head of maternity at each health centre will be a point person for this follow-up register. The study coordinator will call each point person to check if a C-section patient showed up at any health centre. If so, the study coordinator will visit the health centres that patients returned to. During the visit, the study coordinator will refer to the follow-up register to record into REDCap which date the patient returned, wound status, diagnosis, treatment provided and if they were referred to KDH for further care. There will be a similar patient tracker log in the maternity ward reception at KDH to document patients referred to the hospital. This log will be completed by the reception nurse who will notify the study data collector who will input to REDCap. Finally, all patients with phone numbers provided will be called on POD 30 to inquire about any readmissions or visits to other healthcare facilities. Study staff will extract from the clinical chart the presence of an SSI, severity, treatment obtained, need for operative intervention, hospitalisation and/or complications.

## Analyses

All analyses will be completed as intention to treat. For objective 1, the primary outcome is whether a patient returns to care at a health centre or district hospital with a provider-confirmed SSI. We will compare the proportion of patients who returned for follow-up with an SSI in arms 1 and 2 with arm 3 using a two-sided, two-sample test of proportions at the $\alpha=0.05$ significance level. The analyses assume that the rates of true SSIs are constant across the three arms, but that the proportion of these infections that return to care will vary across the study arms as a result of the intervention. We have purposely chosen not to trace patients to establish their true SSI status, as this would interfere with care seeking behaviour. However, we will perform a sensitivity analysis (changing the null hypothesis from $p_1=p_2$ to $p_1=kp_2$, where $k$ reflects differences in SSI rates) to determine under what range of SSI rates the results are still valid. As a secondary outcome for objective 1, we will look at time to return-to-care for patients with SSI dichotomised as within 15 PODs or >15 PODs. We will use a logistic regression model to assess the impact of study arm on timely return to care, controlling for potential confounders collected at enrolment. For objective 2, we will assess the implementation feasibility of the CHW-mHealth intervention by quantifying intervention indicators. For each indicator, we will report the per cent of eligible encounters for which that step was successfully completed, and will categorise a specific

**Table 2** Sample size calculation

|  | Arm 1: home visit+protocol | Arm 2: phone call+protocol | Arm 3: standard of care |
|---|---|---|---|
| Total patients | 364 | 364 | 364 |
| Anticipated SSIs | 55 | 55 | 55 |
| Hypothesised patients to return with SSIs | 44 (80%) | 44 (80%) | 22 (40%) |
| Overall hypothesised proportion that will return with SSI | 0.12 | 0.12 | 0.06 |

SSI, surgical site infection.

component as feasible if at least 85% of eligible counters have that step completed. For arms 1 and 2, we will calculate a comprehensive feasibility measure that will assess the per cent of encounters that successfully implemented the full intervention, which we aim to achieve with at least 85% of patient encounters.

### Power calculation

Over the 12-month study period, we expect 78 patients per month or 1092 patients total to be eligible for inclusion. Assuming a 1-1-1 randomisation, 26 patients per month or 364 patients total will be randomised to each arm (table 2).

We assume a constant SSI rate across the three arms of 15% (based on data from preliminary chart reviews prior to this study, and prior to the first phase of this study which identified the 10.3% prevalence over a 7-month enrolment window). We assume more patients with SSIs will return to care in arms 1 and 2 compared with arm 3 (80% of SSIs in arms 1 and 2 compared with 40% in arm 3). This corresponds to an overall return to care rate of 12% in arms 1 and 2 and 6% in arm 3. We would have an 81% power to detect a difference between the proportion of patients that returned with an SSI in arms 1 and 2 (12%) as compared with arm 3 (6%) with a two-sided test at the $\alpha=0.05$ significance level.

### Patient and public involvement

Patients and/or the public were not involved with the development of the research question or study design. The results of the study however will be disseminated at a community event at the hospital following the completion of the trial.

### ETHICS AND DISSEMINATION

Study participants will be informed on the intent of the study, potential benefits and risks of their enrolment and how these will be minimised. Those who wish to enrol will be informed of their right to withdraw throughout the study period. All data collectors will sign confidentiality training and agreements; study coordinators and CHWs. Risks to privacy will be minimised by having all mobile devices and computers password protected. Data will be stored on a Health Insurance Portability and Accountability Act (HIPAA) compliant servers, and data will be deidentified prior to any analysis.

### Benefits, risks and limitations

The study does not alter the standard of care in any way and therefore there is minimal to no increased risk to the patient. Participants will likely benefit from this study in that the intervention we hypothesise will lead to a timelier diagnosis of SSI and will encourage patients to return to care, which is likely to correlate with improved health outcomes. However, one limitation of this study is that we do not measure health outcomes directly. Patients enrolled in both arms 1 and 2 will have additional contact with a healthcare provider (CHW) beyond the current standard of care. While not all participants may need this earlier screening, as not all will have surgical complications, the risks and discomforts associated with the screening are minimal. Given that patients will be randomised to all three arms, there is a risk of cross-contamination between patients from the same village. However, with our total sample size of 1200 patients, and that Kirehe District has approximately 612 villages with the population relatively evenly distributed, we do not expect more than two to five women per village to be enrolled. Since enrolment will be over 12 months, we expected that this contamination bias will be minimal.

On a systems level, this study will benefit the local providers and research staff to understand whether CHWs can be used in this capacity for postoperative follow-up. If we find that routine follow-up of patients with a CHW (either by phone or in-person visits) leads to a statistically significant higher identification of patients with an SSI, we will then be able to advocate for the use of CHWs for postoperative patients as that currently is not the standard. Furthermore, given the relationship that the study staff has with the CHW coordinator for Kirehe District, KDH, as well as the Ministry of Health, it could lead to a new standard of care for all patients to have regular follow-up after C-section. In addition, this study tracks feasibility indicators, which will inform broader conversations about whether such follow-up is possible in this and similar contexts; this is particularly novel for the arm 2, given that no programmes have used phone calls for postoperative follow-up in the rural areas in the region.

A potential risk will be decreasing the likelihood of a patient return to care when needed under the mHealth-CHW interventions. It is possible that the CHW will give the wrong SSI diagnosis or that a patient may delay return to care because of an expected visit from

a CHW. This risk is moderate and will be monitored by a Data and Safety Monitoring Board (DSMB). Finally, a potential risk would be a breach in confidentiality, resulting in the disclosure of patient information. This risk is considered minimal as unique codes will be used in place of participant names throughout the study. Only principal investigators and study coordinators will have access to the final deidentified database.

## Data and Safety Monitoring Board

The DSMB will be designated to oversee the safety and effectiveness of the study. This committee will include one global surgery expert, one Rwandan health practitioner and one statistician. Meetings of the DSMB will be held twice—once at the start of the study and 6 months after the start of phase II. At the first meeting, the DSMB will discuss the protocol, suggest modifications and establish guidelines to study monitoring by the Board.

At the second meeting, we will present the DSMB an interim analysis report, which will compare rates of return between the three study arms and include a list of adverse events of this study, if any. We anticipate half of the total cohort of patients will be included in this interim analysis. If the proportion who have returned in arms 1 and 2 is significantly lower compared with standard of care, then the study will be stopped or one study arm will be dropped. Furthermore, if there are significantly more complex cases at return (higher rates of readmission or reoperation) in arms 1 or 2, then the study will be stopped or one study arm will be dropped. The outcome of the DSMB review will be summarised in a letter to the institutional review boards (IRBs) of all participating institutions. A recommendation by the DSMB to terminate the study would be communicated to the National Institutes of Health Director, who will then accept or decline the recommendation.

## Ethics approvals

The study has received IRB approval both in the USA and in Rwanda. Any proposed protocol amendments would undergo review and approvals by IRBs before further implementation.

## Dissemination

Results will be disseminated to the staff at KDH, the Rwanda Ministry of Health, including the electronic Health and CHW departments, the Rwanda Surgical Society and PIH. Results will also be disseminated at regional and international conferences and via peer-reviewed publications.

**Contributors** BLH-G, RR and FK are the three primary investigators for the study, instigated the original idea for the study, developed the funding proposal and applied for funding. EG, CH, AM, EN, GN advised the study objectives and scientific content of this study. KAS, TN and RK are study coordinators and contributed to writing of the protocol. MG contributed to writing of the protocol and figures/tables. All authors read and commented on drafts, and approved the final version.

**Funding** This work was supported by The National Institute of Health grant number NIH Grant 1R21EB022369–01.

**Disclaimer** The funders have no role in the study design, collection, management, analysis or interpretation of results.

**Competing interests** None declared.

**Patient consent** Obtained.

**Ethics approval** IRB approval in the USA has been achieved through Partners Healthcare (2016P001943/MGH). The Rwanda National Ethics Committee has reviewed and approved the study (848/RNEC/2016).

**Provenance and peer review** Not commissioned; externally peer reviewed.

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
