## [Reviewer comments · BMJ Open]

ARTICLE DETAILS

TITLE (PROVISIONAL)	Using mobile health technology and community health workers to identify and refer cesarean-related surgical site infections in rural Rwanda: A randomized-control trial protocol
AUTHORS	Sonderman, Kristin; Nkurunziza, Theoneste; Kateera, Fredrick; Gruendl, Magdalena; Koch, Rachel; Gaju, Erick; Habiyakare, Caste; Matousek, Alexi; Nahimana, Evrard; Ntakiyiruta, Georges; Riviello, Robert; Hedt-Gauthier, Bethany

VERSION 1 – REVIEW

REVIEWER	Joseph Forrester Stanford University
REVIEW RETURNED	27-Feb-2018

GENERAL COMMENTS	Well thought out protocol. Screening tool has been validated before. Progressive, in that it engages community health workers.
--

REVIEWER	Pratap Kumar Senior Lecturer, Institute of Healthcare Management, Strathmore Business School, Kenya
REVIEW RETURNED	06-Mar-2018

GENERAL COMMENTS	1. Potential for contamination between arms: The CHW Arm (Arm 1) will require the hired CHW to liaise with the local CHW, find the mother's house, meet the mother, ask three questions, and take a picture of the surgical site. In small, rural communities such as where this study is being conducted I think it likely that knowledge of the intervention would spread to the local CHW and to those in Arm 3. The intervention is relatively simple, and it would be easy, especially for mothers in Arm 1, to share their experience with others, resulting in higher rates of referral overall, and low-than-expected differences in proportions returning between groups. With 81% power of detecting a doubling in rates of return, spread of information between arms could easily affect the statistical significance of the results. Could the authors share why a cluster randomised design might not be better suited for this study? 2. Disparity between SSI incidence at KDH and estimates in the study design: On page 6, line 24, the authors say they "found a 10.3% SSI incidence." But for the statistics (page 14, line 26) they use "assume a constant SSI rate" of 15%. A lower SSI incidence could again lower the significance of the results. Could the authors clarify the disparity? 3. 'Delayed return to care' not defined by health outcomes: As the
---

	authors write, "For patients who develop an SSI, failure to return or a delayed return to care is linked with poor health outcomes." However it is difficult to define what is 'delayed' purely in number of days. In my understanding, a mother in Arm 3 who develops an SSI on POD 5 and presents at a health centre on POD 25 is likely to suffer from poor health outcomes, but would be classified as 'returned to care' w.r.t. the primary outcome. It would, I recognise, be reflected in the secondary objective, which looks at returns before and after POD 15. Given the regular visits made to the health centres, and efforts to extract information from the follow-up register and clinical chart, I wonder if it would be worth considering including health outcomes in the objectives. At the very least, the inability to relate the results to outcomes should be reported as a limitation of the study. 4. Justification of an RCT to ascertain rates of return: The authors state that the "greatest strength is that this is a prospective randomized control trial to most effectively evaluate the impact of a mobile health and CHW intervention on return to care following surgery." The value of returning for follow-up visits following surgery is clear. Whether the follow-up is more effective when initiated by phone or home visit by a CHW is a valid question, but probably driven more by the health system context and relative costs than the outcome of an RCT. In a setting like Rwanda, where CHWs are likely to be used more widely and for many tasks, training them to ask three questions for SSI is unlikely to add significantly to training costs. If CHW coverage is not high, then establishing a home screening program for SSI by CHWs is unlikely to be cost-effective; phone-based screening would, quite conceivably, provide an alternative at low cost. The conduct of an RCT should be justified by the likely scenarios that the results of the study would engender. I would like the authors to include the likely impact of the proposed RCT on policy around using CHWs for SSI screening.
--	--

VERSION 1 – AUTHOR RESPONSE

Reviewer: 1

Well thought out protocol. Screening tool has been validated before. Progressive, in that it engages community health workers.

Response: Thank you, we appreciate your comments.

Reviewer: 2

1. Potential for contamination between arms: The CHW Arm (Arm 1) will require the hired CHW to liaise with the local CHW, find the mother's house, meet the mother, ask three questions, and take a picture of the surgical site. In small, rural communities such as where this study is being conducted I think it likely that knowledge of the intervention would spread to the local CHW and to those in Arm 3. The intervention is relatively simple, and it would be easy, especially for mothers in Arm 1, to share their experience with others, resulting in higher rates of referral overall, and low-than-expected differences in proportions

returning between groups. With 81% power of detecting a doubling in rates of return, spread of information between arms could easily affect the statistical significance of the results. Could the authors share why a cluster randomized design might not be better suited for this study?

Tel: 617-525-7300 · Fax: 617-525-7723

One Brigham Circle · 1620 Tremont Street, 4th floor · Boston, MA 02120

Page 1

Response: Thank you for your thoughtful comment. We agree that there is the risk of cross contamination with the current design and that this could bias the results towards the null. A cluster design, however, would require a considerably larger number of patients for adequate power if there is clustering in behaviors, which is not feasible under the R21 grant mechanism which is intended to be exploratory for bigger bodies of research. Further, designing such a study would require knowledge of the clustering of the outcome, by intervention, which we did not have a priori. Given that the total sample size is about 1200 patients, and that there are approximately 612 villages in Kirehe District, with the population relatively evenly distributed, we do not expect more than 2-5 women per village to be enrolled. Since this enrollment will be over 12 months, we expect that this contamination bias will be minimal. We have modified our Benefits, Risks and Limitations section to address this:

Benefits, Risks and Limitation (page 14): Given that patients will be randomized to all three arms, there is a risk of cross contamination between patients from the same village. However, with our total sample size of 1200 patients, and that Kirehe District has approximately 612 villages with the population relatively evenly distributed, we do not expect more than 2-5 women per village to be enrolled. Since enrollment will be over 12 months, we expected that this contamination bias will be minimal.

2. Disparity between SSI incidence at KDH and estimates in the study design: On page 6, line 24, the authors say they "found a 10.3% SSI incidence." But for the statistics (page 14, line 26) they use "assume a constant SSI rate" of 15%. A lower SSI incidence could again lower the significance of the results. Could the authors clarify the disparity?

Response: Thank you for this comment and we agree this needs further clarification. The protocol for this study that was approved for the R21 grant was written prior to the first phase of this study which included the validation of the SSI screening tool as well as the outcome of a 10.3% identified SSI incidence. The 15% incidence of SSI was determined by paper chart review in preparation for this entire project and protocol. Therefore, we are using the 15% incidence as the 10.3% is a preliminary outcome of this study. This SSI rate is an estimate, and given that the rate can vary over a year, we still believe that our preliminary estimate is reasonable and consistent with the literature. This has been further clarified in the Power Calculation section:

Power calculation (page 13): We assume a constant SSI rate across the three arms of 15% (based on data from preliminary chart reviews prior to this study, and prior to the first phase of this study which identified the 10.3% prevalence over a seven-month enrollment window).

Tel: 617-525-7300 · Fax: 617-525-7723

One Brigham Circle · 1620 Tremont Street, 4th floor · Boston, MA 02120

3. 'Delayed return to care' not defined by health outcomes: As the authors write, "For patients who develop an SSI, failure to return or a delayed return to care is linked with poor health outcomes." However it is difficult to define what is 'delayed' purely in number of days. In my understanding, a mother in Arm 3 who develops an SSI on POD 5 and presents at a health centre on POD 25 is likely to suffer from poor health outcomes, but would be classified as 'returned to care' w.r.t. the primary outcome. It would, I recognise, be reflected in the secondary objective, which looks at returns before and after POD 15. Given the regular visits made to the health centres, and efforts to extract information from the follow-up register and clinical chart, I wonder if it would be worth considering including health outcomes in the objectives. At the very least, the inability to relate the results to outcomes should be reported as a limitation of the study.

Response: We agree that there are limitations of having the primary outcome be a metric of healthcare utilization, rather than a health outcome. However, we believe that delay in presentation to a provider for a serious problem such as SSI is a sensitive and important stand-alone endpoint. We have added to the Benefit, Risks, and Limitations section the distinction between healthcare utilization and health outcomes.

Benefits, Risks and Limitations (page 14): Participants will likely benefit from this study in that the intervention we hypothesize will lead to a timelier diagnosis of SSI and will encourage patients to return to care, which is likely to correlate with improved health outcomes. However, one limitation of this study is that we do not measure health outcomes directly.

4. Justification of an RCT to ascertain rates of return: The authors state that the "greatest strength is that this is a prospective randomized control trial to most effectively evaluate the impact of a mobile health and CHW intervention on return to care following surgery." The value of returning for follow-up visits following surgery is clear. Whether the follow-up is more effective when initiated by phone or home visit by a CHW is a valid question, but probably driven more by the health system context and relative costs than the outcome of an RCT. In a setting like Rwanda, where CHWs are likely to be used more widely and for many tasks, training them to ask three questions for SSI is unlikely to add significantly to training costs. If CHW coverage is not high, then establishing a home screening program for SSI by CHWs is unlikely to be cost-effective; phone-based screening would, quite conceivably, provide an alternative at low cost. The conduct of an RCT should be justified by the likely scenarios that the results of the study would engender. I would like the authors to include the likely impact of the proposed RCT on policy around using CHWs for SSI screening.

Tel: 617-525-7300 · Fax: 617-525-7723

One Brigham Circle · 1620 Tremont Street, 4th floor · Boston, MA 02120

Response: We agree that the potential benefits have not been adequately described and have amended the Benefits, Risks, and Limitation section to expand on the potential impact of this study on the care of patients:

Benefits, Risks and Limitations (page 14): On a systems level, this study will benefit the local providers and research staff to understand whether CHWs can be used in this capacity for postoperative follow-up. If we find that routine follow up of patients with a CHW (either by phone or in-person visits) leads to a statistically significant higher identification of patients with an SSI, we will then be able to advocate for the use of CHWs for postoperative patients as that currently is not the standard. Further, given the relationship that the study staff has with the CHW coordinator for Kirehe District, KDH, as well as the Ministry of Health, it could lead to a new standard of care for all patients to have regular follow up after cesarean section. In addition, this study tracks feasibility indicators, which will inform broader conversations about whether such follow-up is possible in this and similar contexts; this is particularly novel for the Arm 2, given that no programs have used phone calls for post-operative follow-up in the rural areas in the region

VERSION 2 – REVIEW

REVIEWER	Pratap Kumar Institute of Healthcare Management, Strathmore Business School
REVIEW RETURNED	02-Apr-2018
GENERAL COMMENTS	The authors have satisfactorily addressed the comments made in the earlier review.